

# Non-obese non-alcoholic fatty liver disease and the risk of chronic kidney disease: a systematic review and meta-analysis

Yixian You*, Xiong Pei*, Wei Jiang, Qingmin Zeng, Lang Bai, Taoyou Zhou, Xiaoju Lv, Hong Tang and Dongbo Wu

Center of Infectious Diseases, West China Hospital, Sichuan University, Chengdu, China
* These authors contributed equally to this work.

## ABSTRACT

**Background:** Data on risk of developing chronic kidney disease (CKD) between non-obese and obese non-alcoholic fatty liver disease (NAFLD) patients are limited. We aimed to reveal the risk difference of incident CKD between non-obese and obese NAFLD patients.

**Methods:** We searched PubMed, Embase, and Web of Science databases for studies which reported the incidence of CKD in non-obese and obese NAFLD from inception to 10 March 2024. The primary and secondary outcomes were pooled. Subgroup analysis was used to examine the heterogeneity.

**Results:** A total of 15 studies were incorporated. The incidence of CKD in non-obese and obese NAFLD were 1,450/38,720 (3.74%) and 3,067/84,154 (3.64%), respectively. Non-obese NAFLD patients had a comparable risk of CKD as obese NAFLD (odds ratio [OR] 0.92, 95% confidence interval [95% CI] [0.72–1.19], $I^2 = 88\%$). No differences in estimated glomerular filtration rate and serum creatinine between non-obese and obese NAFLD were found. The mean differences (MD) and 95% CI were 0.01 [−0.02 to 0.04] and 0.50 [−0.90 to 1.90], respectively. In subgroup analyses, non-obese NAFLD had higher eGFR when diagnosed with ultrasound (MD 1.45, 95% CI [0.11–2.79], $I^2 = 21\%$). Non-obese NAFLD had higher creatinine in non-Asian (MD 0.06, 95% CI [0.01–0.11], $I^2 = 55\%$) and when taking BMI > 30 as the criterion for obesity (MD 0.06, 95% CI [0.00–0.12], $I^2 = 76\%$). The occurrence of CKD did not differ when non-obese NAFLD were categorized into overweight and normal-weight types.

**Conclusions:** Non-obese NAFLD patients experienced the same risk of CKD compared to obese NAFLD.

## INTRODUCTION

Nonalcoholic fatty liver disease (NAFLD), characterized by accumulated fat in liver cells, has emerged as the most common causes of chronic liver disease (CLD), impacting an estimated 25% of the population worldwide and 30% of population in China (*Wong, Ting & Chan, 2018*; *Younossi et al., 2019, 2016*). NAFLD carries a substantially elevated risk of

Corresponding authors
Hong Tang,
htang6198@hotmail.com
Dongbo Wu,
dongbohuaxi@scu.edu.cn

systemic complications, including cardiovascular disease (*Tang et al., 2022*; *Toh et al., 2022*), extrahepatic and hepatic malignancy (*Tan et al., 2022*), and chronic kidney disease (CKD) (*Byrne & Targher, 2020*). The reciprocal relationships between NAFLD and obesity has been continuously reported (*Byrne & Targher, 2015*; *Del Ben et al., 2014*). In this context, most NAFLD subjects are overweight/obese with components of metabolic syndrome (*Fazel et al., 2016*; *Yki-Järvinen, 2014*). Despite a strong association with obesity, increasing data indicate that a proportion of subjects with NAFLD are lean or non-obese (*Golabi et al., 2019*; *Younossi, 2017*).

Lean NAFLD was initially reported in Asian populations, which refers NAFLD occur in individuals who are not obese and have a normal body mass index (BMI) (*Younes & Bugianesi, 2019*; *Young et al., 2020*). Global prevalence of lean NAFLD has been estimated to range from 5.1% to 11.2% in the general population with the highest prevalence in Asia, while the global prevalence of lean individuals in NAFLD patients varies from 19.2% to 25.3% (*Ye et al., 2020*; *Young et al., 2020*). Despite a better metabolic condition, the impact of NAFLD on lean individuals may be more severe, with a greater incidence of advanced fibrosis, cardiovascular events, and liver-related deaths (*Younes et al., 2022*). NAFLD and CKD are well linked since NAFLD is frequently accompanied by risk factors for CKD, such as visceral obesity, type 2 diabetes, insulin resistance, and metabolic syndrome. Several studies have been reported that NAFLD increases the incidence of CKD (*Arase et al., 2011*; *Park et al., 2019*; *Sinn et al., 2017*; *Zhou et al., 2023*). However, the incidence of CKD in non-obese NAFLD patients compared with obese NAFLD remains unclear. Consistent conclusions regarding whether non-obese patients with NAFLD on the risk of developing CKD than obese patients are still not drawn (*Akahane et al., 2020*; *Das et al., 2010*). In this study, we performed a meta-analysis of relevant studies to investigate whether non-obese NAFLD is associated with a greater incidence of CKD compared with obese NAFLD.

## METHOD

### Data sources and search strategy

The protocol for this systematic review was registered in advance with the International Prospective Register of Systematic Reviews (PROSPERO) (no. CRD42023442546). We systematically searched PubMed, Embase and Web of Science databases from the inception of the databases until 10 March 2024 to determine studies exploring the association between non-obese NAFLD and incidence of CKD for keywords and MeSH terms synonymous with 'non-alcoholic fatty liver', 'non-obese', and 'chronic kidney disease'. The full search strategy was provided in Table S1. Additionally, the reference lists of previous publications will be thoroughly reviewed to identify articles not captured in the initial search, and PubMed's "related articles" feature will be used to find other potentially relevant studies.

### Study selection
#### Inclusion criteria

Studies that satisfy all of the following criteria will be considered for inclusion: (1) studies were published in English and full text are available; (2) studies (prospective, retrospective,

and cross-sectional) examined the association the risk of CKD between non-obese and obese NAFLD or between overweight and lean NAFLD; (3) adults 18 years of age or older; (4) NAFLD was diagnosed using imaging, histology, or noninvasive scoring methods, such as the FIB-4 score, fatty liver index, or NAFLD liver fat score, while excluding other causes of hepatic steatosis; (5) CKD was defined by estimated glomerular filtration rate (eGFR) <60 ml/min/1.73 m$^2$, as estimated using the Chronic Kidney Disease Epidemiology Collaboration (CKD-EPI) equations or studies reported the serum creatinine or eGFR.

*Exclusion criteria*

The exclusion criteria are as follows: (1) congress abstracts, practice guidelines, reviews, theses, editorials, case reports, non-human studies; (2) studies that did not exclude heavy drinkers and other potential risks of hepatic steatosis; (3) studies which did not report ORs and 95% CIs or events for the outcomes of interest; (4) studies conducted in the pediatric population (<18 years); and (5) coexisting commonly occurring chronic liver diseases and potential contributors to steatosis, such as over consumption of alcohol and infection with hepatitis viruses, are explicitly excluded.

## Outcome measures

The major outcome indicator was the occurrence of incident CKD among individuals with non-obese NAFLD compared with the incidence of CKD among those with obese NAFLD. Serum creatinine, and eGFR were taken as the secondary outcomes. Risk of primary outcomes and secondary outcomes in obese and non-obese patients with NAFLD were compared.

## Data extraction and quality assessment

For each study, data were extracted independently by two authors (XP, YXY). The name of the first author, publication year, region of the study population, type of study, number of participants, diagnostic method of NAFLD, diagnostic of incident CKD, the definition of obese/lean, events, OR, 95% CI, and adjusted confounders were extracted. Two authors (WJ and QMZ) assessed the risk of bias independently. When disagreements arise, a more experienced author are consulted (HT). The Newcastle Ottawa Scale (NOS) (*Stang, 2010*) was used to judge quality of prospective and retrospective studies, and The Agency for Health Care Research and Quality (AHRQ) was used to judge quality of cross-sectional studies as recommended by the Cochrane Collaboration. The NOS scale evaluates studies using a star rating system (up to nine stars) in three ways: participant selection, study group comparability, and related outcome determination. Studies that received seven or more stars were judged to be at low risk of bias, those that received six stars were judged to be at moderate risk of bias, and those that received less than six stars were judged to be at high risk of bias. The AHRQ checklist consists of 11 items, with response options of "yes", "no", and "not sure". Scores between 0 and 3 are classified as low quality, 4 to 6 as moderate quality, and 7 to 11 as high quality. Eligible study results were pooled, and overall estimated effect sizes were derived using a random effects model, since this approach considers any differences between studies, even in the absence of statistically significant heterogeneity. Publication bias was assessed when the number of included studies

exceeded 10. Sensitivity analysis was performed using random-effects models to pool the results of the subgroups.

### Data synthesis and analysis

For dichotomous variable, the ORs and 95% CIs were taken as the effect size. If the study only reported the number of outcomes of interest for non-obese and obese NAFLD groups, ORs were calculated using Revman 5.3. For continuous variables, the mean ± standard deviation (SD) was extracted. If studies reported ORs with different covariate adjustment, the ORs that indicated the greatest degree of adjustment for potential confounders were extracted. Before pooling adjusted OR, OR should be transformed to log-OR. The effect sizes of all eligible studies were pooled using a random-effects model.

### Clinical definition

The WHO and Asian BMI thresholds vary owing to differences in health risks and body composition in specific populations. Obesity is defined in the WHO standards as more than 30 kg/m$^2$, overweight as 25–29.9 kg/m$^2$, normal weight as 18.5–24.9 kg/m$^2$, and underweight as less than 18.5 kg/m$^2$. The Asian Standard defines obese as 25 kg/m$^2$, overweight as 23–24.9 kg/m$^2$, normal weight as 18.5–22.9 kg/m$^2$, and underweight as less than 18.5 kg/m$^2$. In our study, Asian criteria were applied to all Asian populations, while for non-Asian populations, WHO criteria were applied. Non-obese comprise overweight and normal individuals. The lean NAFLD referred to individual with normal weight or bellow.

## Statistical analyses

Risk estimates were pooled using a random-effects model. The I$^2$ statistic and its 95% CI were calculated to assess statistical heterogeneity across studies: 0% no heterogeneity, 0–25% very low heterogeneity, 25–50% low heterogeneity, 50–75% moderate heterogeneity, and 75–100% high heterogeneity. In case of I$^2$ values >50%, we conducted stratification analyses by study country, methodology used for the diagnosis of NAFLD (ultrasonography, liver biopsy, and non-invasive fibrosis scores), BMI cutoff value, and study quality. Sensitivity analyses were carried out by replicating the meta-analysis after excluding one study at a time to assess whether any of the studies had a significant impact on the pooled estimates. If there were multiple subgroups in a study, the data are combined. Serum albumin was expressed in g/dL, serum creatine was expressed in mg/dL. $P \leq 0.05$ was taken as the significance setting. Kappa statistics was used to reveal the difference between the authors during data screening and selection. All meta-analyses were conducted with the help of RevMan 5.3 and Stata 16.

## RESULT

### Characteristics of included studies

Initially, 1,677 articles were obtained from the databases. After removing 51 duplicated articles, 1,626 publications remained for the abstract selection. A total of 1,611 studies were excluded for the stated reasons in the flowchart (Fig. 1). Retrieving the reference lists of all relevant articles, one study were found. A total of 15 studies were included into the meta-analysis finally (*Akahane et al., 2020*; *Ampuero et al., 2018*; *Chon et al., 2020*; *Fan*

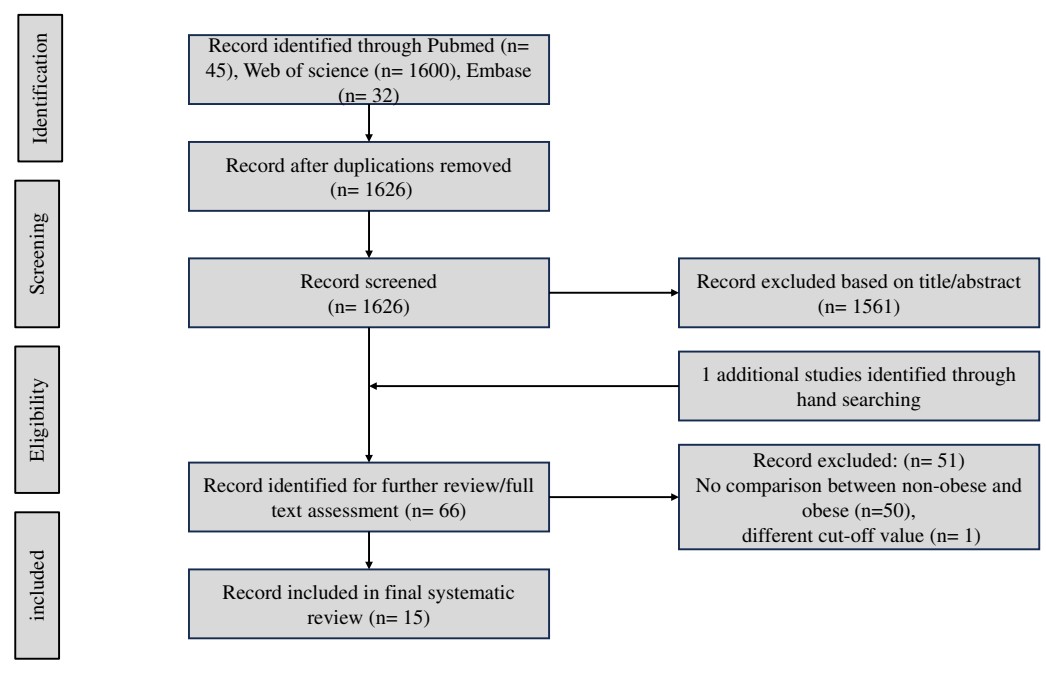

**Figure 1 Workflow diagram of study selection.**

*et al., 2023; Hu et al., 2022a, 2022b; Iwaki et al., 2022; Kim et al., 2022; Kwon et al., 2023; Leung et al., 2017; Liu et al., 2021; Mikolasevic et al., 2020; Nabi et al., 2023; Ragab et al., 2021; Yang et al., 2018*).

As summarized in Table S2, the studies include four cross-sectional (*Akahane et al., 2020; Ampuero et al., 2018; Hu et al., 2022a; Kim et al., 2022*), three retrospective (*Iwaki et al., 2022; Kim et al., 2022; Yang et al., 2018*) and eight prospective (*Chon et al., 2020; Fan et al., 2023; Hu et al., 2022b; Leung et al., 2017; Liu et al., 2021; Mikolasevic et al., 2020; Nabi et al., 2023; Ragab et al., 2021*) cohort studies. NAFLD was diagnosed by ultrasound (*Akahane et al., 2020; Hu et al., 2022a; Kwon et al., 2023; Mikolasevic et al., 2020; Ragab et al., 2021; Yang et al., 2018*) (6/15), liver biopsy (*Ampuero et al., 2018; Iwaki et al., 2022; Leung et al., 2017*) (3/15), non-invasive scores (*Chon et al., 2020; Fan et al., 2023; Kim et al., 2022; Liu et al., 2021; Nabi et al., 2023*) (FLI: 3/15, LFS: 2/15, FIB-4: 1/15), and one was unclear (*Hu et al., 2022b*) (1/15). A total of 9/15 of these studies were conducted in Asian (four in China (*Hu et al., 2022a, 2022b; Leung et al., 2017; Yang et al., 2018*), four in Korea (*Chon et al., 2020; Iwaki et al., 2022; Kim et al., 2022; Kwon et al., 2023*), and one in Japan (*Akahane et al., 2020*), respectively), and 6/15 of these studies were conducted in non-Asian (two in UK Biobank (*Fan et al., 2023; Liu et al., 2021*), one in France (*Nabi et al., 2023*), one in Egypt (*Ragab et al., 2021*), one in Croatia (*Mikolasevic et al., 2020*), and one in Spain (*Ampuero et al., 2018*), respectively). As two studies were included in the population inside UK Biobank, to avoid duplication of populations, we only included the study with the larger study population in the initial data integration. Because the other study subdivided non-obese NAFLD into overweight and normal weight types, we included this study in the subgroup analysis. For non-obese/obese NAFLD, studies used different thresholds as follows: BMI > 23 kg/m$^2$ (1/15) (*Hu et al., 2022a*), BMI > 25 kg/m$^2$
| Study or Subgroup | log[Odds Ratio] | SE | Weight | Odds Ratio IV, Random, 95% CI |
|---|---|---|---|---|
| Akahane et al. 2020 Japan | -0.7438 | 0.2089 | 11.9% | 0.48 [0.32, 0.72] |
| Chon et al. 2020 Korea | -0.1203 | 0.1254 | 14.3% | 0.89 [0.69, 1.13] |
| Hu et al. 2022a China | 0.5852 | 0.3295 | 8.5% | 1.80 [0.94, 3.42] |
| Hu et al. 2022b China | 0.0797 | 0.0835 | 15.2% | 1.08 [0.92, 1.28] |
| Kwon et al. 2023 Korea | -0.4446 | 0.1999 | 12.1% | 0.64 [0.43, 0.95] |
| Liu et al. 2021 UK Biobank | -0.3716 | 0.0455 | 15.8% | 0.69 [0.63, 0.75] |
| Mikolasevic et al. 2020 Croatia | 0.2858 | 0.2085 | 11.9% | 1.33 [0.88, 2.00] |
| Nabi et al. 2023 France | 0.9123 | 0.262 | 10.3% | 2.49 [1.49, 4.16] |
| | | | | |
| Total (95% CI) | | | 100.0% | 0.97 [0.74, 1.28] |

Heterogeneity: Tau² = 0.12; Chi² = 62.65, df = 7 (P < 0.00001); I² = 89%
Test for overall effect: Z = 0.20 (P = 0.84)

**Figure 2** Forest plot and pooled estimates of the effect of non-obese NAFLD on the risk of incident CKD compared to obese NAFLD (*Akahane et al., 2020*; *Chon et al., 2020*; *Hu et al., 2022a*, *2022b*; *Kwon et al., 2023*; *Liu et al., 2021*; *Mikolasevic et al., 2020*; *Nabi et al., 2023*).

(8/15) (*Akahane et al., 2020*; *Chon et al., 2020*; *Hu et al., 2022b*; *Iwaki et al., 2022*; *Kim et al., 2022*; *Kwon et al., 2023*; *Leung et al., 2017*; *Yang et al., 2018*), BMI > 30 kg/m² (5/15) (*Ampuero et al., 2018*; *Fan et al., 2023*; *Liu et al., 2021*; *Mikolasevic et al., 2020*; *Nabi et al., 2023*), and not clear (1/15) (*Ragab et al., 2021*). Overall, in the 15 studies, there was 1,450 CKD cases in 38,720 non-obese NAFLD (3.74%) while 3,067 CKD cases in 84,154 obese NAFLD (3.64%). Eight studies reported the incident CKD, seven reported serum creatine, five reported eGFR, and four reported the serum albumin between non-obese and obese NAFLD or overweight and lean NAFLD.

Regarding quality assessment, one study received eight stars, three studied received seven stars, six studied received six stars at the NOS, one study received eight stars, two studied received seven stars, two studied received six stars at the AHRQ, indicating that the general quality of the included studies was moderate (Table S1).

## Non-obese NAFLD and risk of incident CKD

Non-obese NAFLD was not significantly associated with an increased risk of incident CKD compared to obese NAFLD (random-effects OR 0.92, 95% CI [0.72–1.19]; I² = 88%) (Fig. 2). When stratified by region (Fig. 3A), the ORs for incident CKD in Asian and non-Asian participants were 0.81 (95% CI [0.60–1.09]) and 1.28 (95% CI [0.59–2.74]), respectively. Stratification by diagnosis method (Fig. 3B) showed ORs of 0.82 (95% CI [0.52–1.28]) for ultrasound-based diagnoses and 1.07 (95% CI [0.65–1.77]) for score-based diagnoses. For study type (Fig. 3C), the ORs were 0.76 (95% CI [0.39–1.48]) for cross-sectional studies and 0.92 (95% CI [0.72–1.19]) for longitudinal studies. When stratified by BMI cut-offs (Fig. 3D), the ORs for CKD were 1.04 (95% CI [0.38–2.84]) for BMI > 23 kg/m², 1.01 (95% CI [0.65–1.56]) for BMI > 25 kg/m², and 0.93 (95% CI [0.49–1.76]) for BMI > 30 kg/m². Finally, when stratified by study quality (Fig. 3E), the ORs were 1.00 (95% CI [0.70–1.44]) for high-quality studies and 0.80 (95% CI [0.54–1.18]) for low-quality studies.

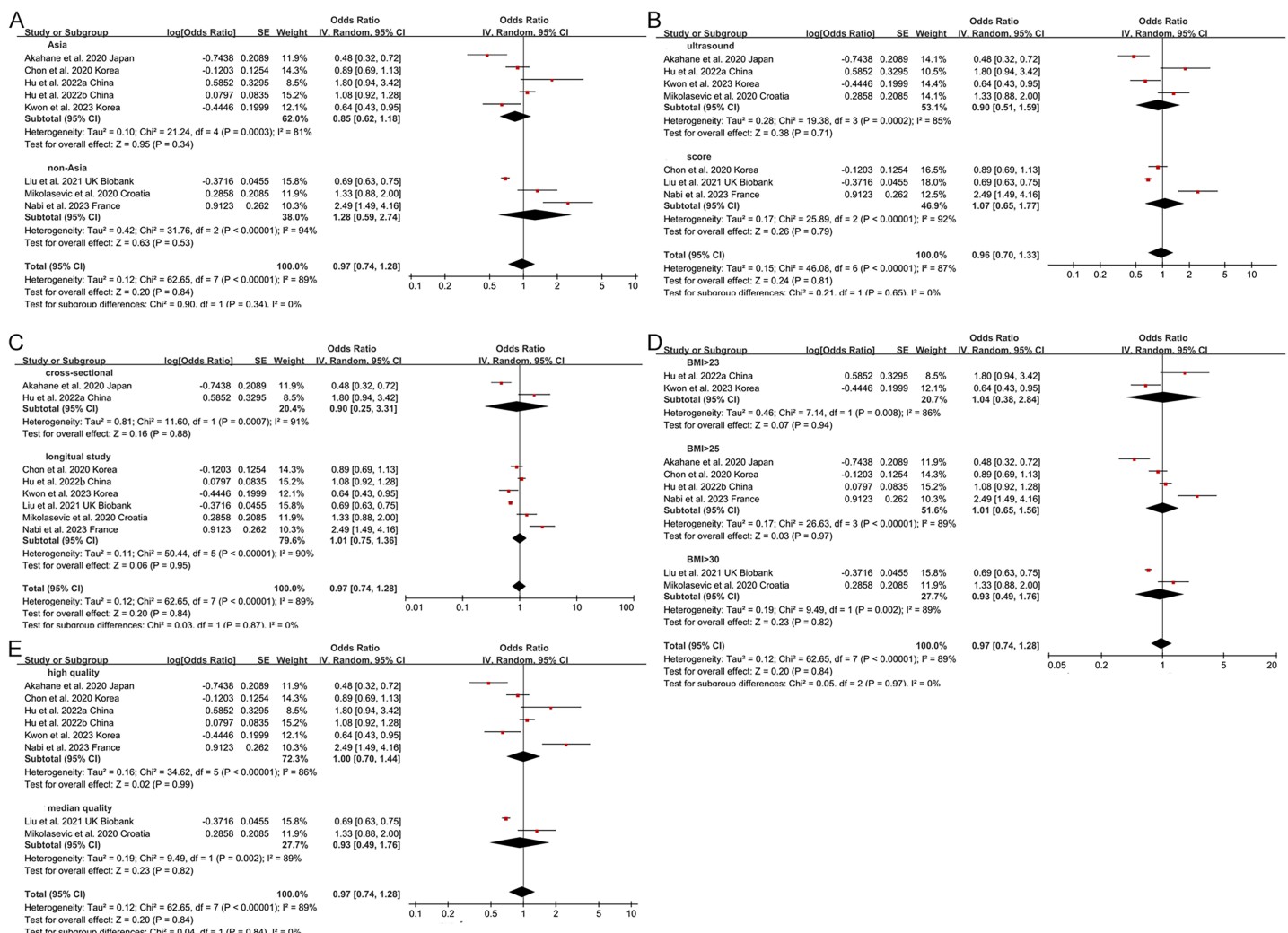

**Figure 3** Forest plot and pooled estimates of the effect of non-obese NAFLD on the risk of incident CKD compared to obese NAFLD stratified by region (A), the methodology used for the diagnosis of NAFLD (B), type of study (C), the cut-off value of non-obese NAFLD (D), quality of studies (E) (*Akahane et al., 2020*; *Chon et al., 2020*; *Hu et al., 2022a*, *2022b*; *Kwon et al., 2023*; *Nabi et al., 2023*; *Iwaki et al., 2022*; *Kim et al., 2022*; *Yang et al., 2018*; *Leung et al., 2017*; *Ampuero et al., 2018*; *Mikolasevic et al., 2020*; *Ragab et al., 2021*).

Moreover, two studies reported the CKD cases in obese, overweight, and lean NAFLD. Then, we analyzed the risk of CKD stratified by BMI. Compared to obese NAFLD individuals, the overweight and lean individuals did not deserve a higher CKD risk, with a pooled OR and 95% CI 0.86 [0.48–1.55], and 1.07 [0.45–2.54], respectively. The lean NAFLD also had a comparable risk of CKD with overweight NAFLD [1.15 (0.89, 1.47)] (Fig. S1).

## The association of non-obese NAFLD with both eGFR and serum creatinine

Non-obese NAFLD was not significantly associated with a lower eGFR compared to the obese NAFLD group (random-effects MD 0.50, 95% CI [−0.90 to 1.90]; $I^2 = 83\%$) (Fig. 4). When stratified by diagnosis method, the pooled MD for ultrasound-based diagnosis was

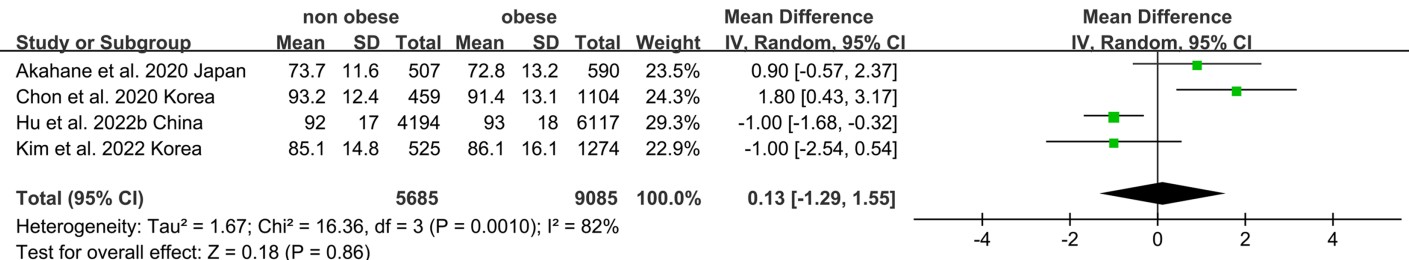

**Figure 4** **Forest plot and pooled estimates of the effect of non-obese NAFLD on the level of eGFR compared to obese NAFLD.** eGFR, estimated glomerular filtration rate (*Akahane et al., 2020*; *Hu et al., 2022b*; *Kim et al., 2022*; *Chon et al., 2020*).

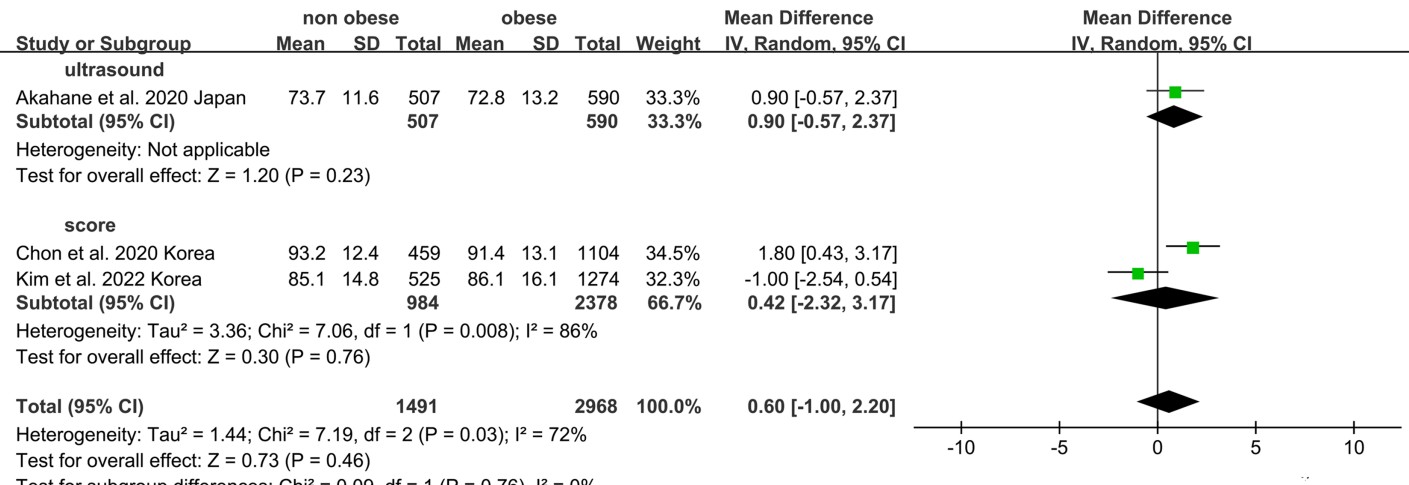

**Figure 5** **Forest plot and pooled estimates of the effect of non-obese NAFLD on the level of eGFR compared to obese NAFLD stratified by the methodology used for the diagnosis of NAFLD** (*Akahane et al., 2020*; *Kim et al., 2022*; *Chon et al., 2020*).

|  | non obese | | | obese | | | | Mean Difference | Mean Difference |
| Study or Subgroup | Mean | SD | Total | Mean | SD | Total | Weight | IV, Random, 95% CI | IV, Random, 95% CI |
| --- | --- | --- | --- | --- | --- | --- | --- | --- | --- |
| Ampuero et al. 2018 Spain | 0.85 | 0.34 | 345 | 0.82 | 0.29 | 713 | 16.5% | 0.03 [−0.01, 0.07] | |
| Iwaki et al. 2022 Japan | 0.67 | 0.2 | 58 | 0.74 | 0.17 | 165 | 13.4% | −0.07 [−0.13, −0.01] | |
| Kim et al. 2022 Korea | 0.9 | 0.2 | 525 | 0.9 | 0.2 | 1274 | 20.6% | 0.00 [−0.02, 0.02] | |
| Leung et al. 2017 China | 0.89 | 0.26 | 72 | 0.84 | 0.25 | 235 | 11.6% | 0.05 [−0.02, 0.12] | |
| Mikolasevic et al. 2020 Croatia | 1 | 0.15 | 251 | 0.91 | 0.27 | 232 | 17.0% | 0.09 [0.05, 0.13] | |
| Ragab et al. 2020 Egypt | 1.12 | 1.57 | 14 | 1.34 | 1.96 | 27 | 0.1% | −0.22 [−1.33, 0.89] | |
| Yang et al. 2018 China | 0.88 | 0.14 | 356 | 0.89 | 0.14 | 470 | 20.7% | −0.01 [−0.03, 0.01] | |
| **Total (95% CI)** | | | **1621** | | | **3116** | **100.0%** | **0.01 [−0.02, 0.05]** | |

Heterogeneity: Tau² = 0.00; Chi² = 30.16, df = 6 (P < 0.0001); I² = 80%
Test for overall effect: Z = 0.84 (P = 0.40)

**Figure 6** **Forest plot and pooled estimates of the effect of non-obese NAFLD on the level of serum creatine compared to obese NAFLD** (*Iwaki et al., 2022*; *Kim et al., 2022*; *Yang et al., 2018*; *Leung et al., 2017*; *Ampuero et al., 2018*; *Mikolasevic et al., 2020*; *Ragab et al., 2021*).

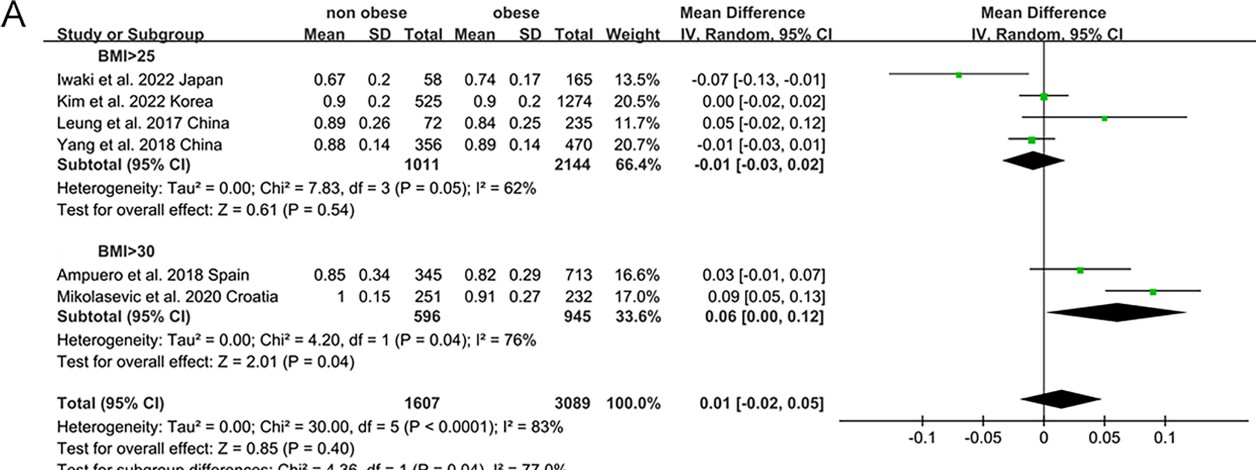

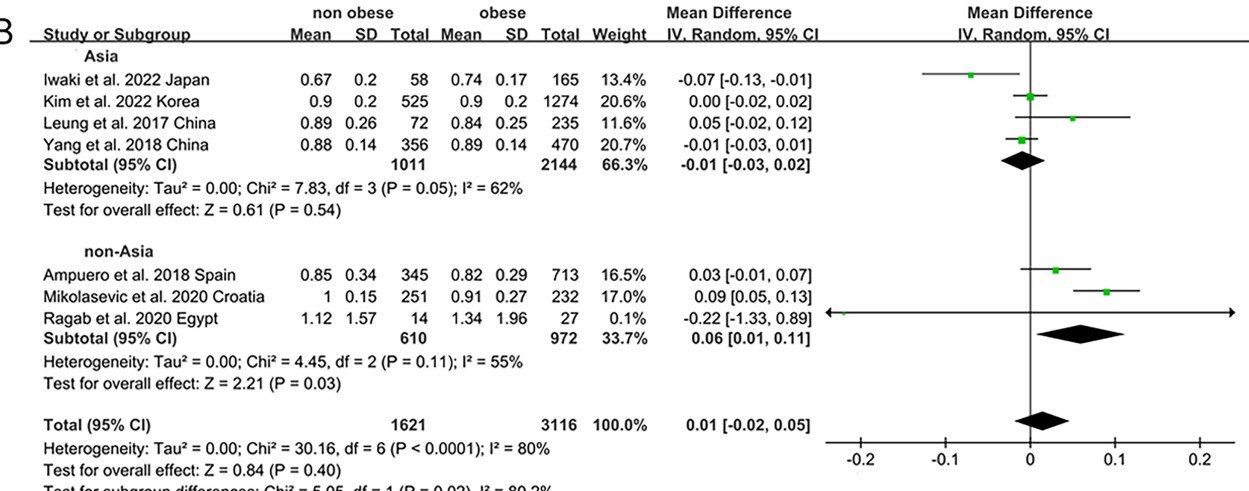

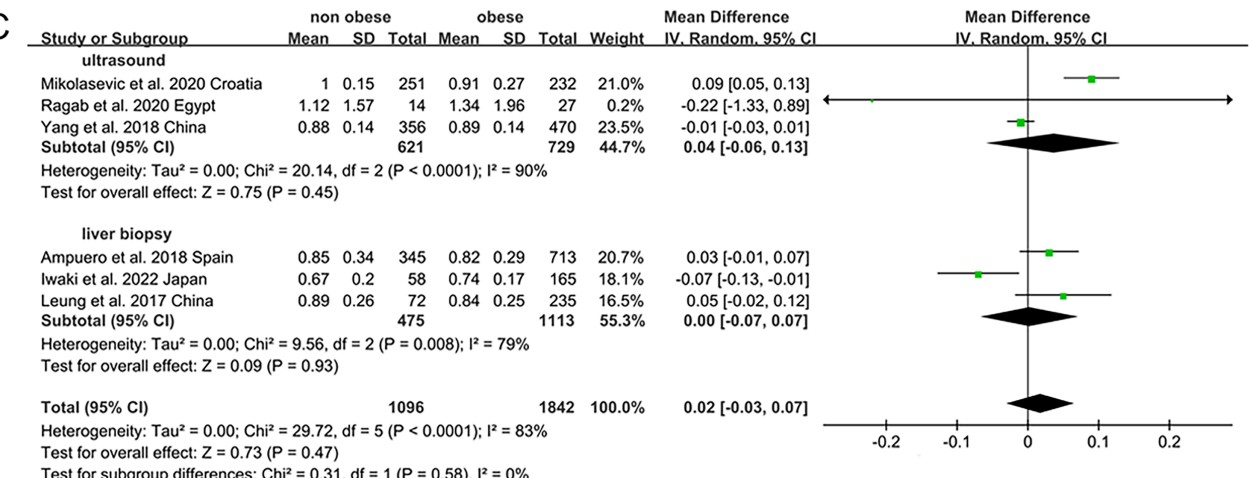

**Figure 7 Forest plot and pooled estimates of the effect of non-obese NAFLD on the level of serum creatine compared to obese NAFLD stratified by the cut-off value of non-obese NAFLD (A), region (B), the methodology used for the diagnosis of NAFLD (C)** (*Iwaki et al., 2022*; *Kim et al., 2022*; *Yang et al., 2018*; *Leung et al., 2017*; *Ampuero et al., 2018*; *Mikolasevic et al., 2020*; *Ragab et al., 2021*).

1.45 (95% CI [0.11–2.79]; $I^2$ = 21%), while for non-invasive score, it was 0.42 (95% CI [−2.32 to 3.17]; $I^2$ = 68%) (Fig. 5). Similarly, non-obese NAFLD was not significantly associated with elevated serum creatinine levels compared to obese NAFLD (random-effects MD 0.01, 95% CI [−0.02 to 0.04]; $I^2$ = 77%) (Fig. 6). Stratified analyses by BMI cut-off, country, and diagnosis method of NAFLD showed no significant differences across subgroups (Fig. 7).

### Exploration of heterogeneity and publication bias

Subgroup analyses were performed by exploring heterogeneity (Figs. 3, 5, 7). It is notable that no association between NAFLD and risk of CKD was observed in any of the subgroups. The results of the meta-analysis were confirmed to be robust through sensitivity analyses (Figs. S2–S4). However, publication bias was not assessed due to the limited number of included studies.

## DISCUSSION

In our study, we aimed to compare the risk of CKD in patients with non-obese NAFLD with that of patients with obese NAFLD. We found no elevated risk of CKD in non-obese and obese NAFLD patients. This result remained consistent even when stratified by BMI, diagnostic methods, region, article type, and quality. When subdividing non-obese NAFLD patients into lean and overweight groups, no difference in CKD occurrence was observed. Additionally, serum creatinine, and eGFR comparisons between non-obese and obese NAFLD patients showed no significant differences.

Non-obese and obese patients with NAFLD share a common altered metabolic profile (*Sookoian & Pirola, 2017*). Despite a better metabolic profile, lean NAFLD was associated with advanced fibrosis (*Nabi et al., 2023*), while obesity itself can cause kidney injury (*Kalaitzidis & Siamopoulos, 2011*; *Mallamaci & Tripepi, 2013*). Thus, the risk of CKD in non-obese NAFLD compared to obese NAFLD remains uncovered. Our meta-analysis suggested that patients with non-obese NAFLD have a comparable risk of developing CKD compared to obese NAFLD (random-effects OR 0.92, 95% CI [0.72–1.19]; $I^2$ = 88%). *Nabi et al. (2023)* argued that lean status was associated with an increased risk of CKD in patients with NAFLD (aHR = 2.49, 95% CI [1.49–4.16]). *Akahane et al. (2020)* revealed that obesity was a risk factor of CKD in NAFLD (OR 2.104, 95% CI [1.397–3.168]). *Hu et al. (2022a)* found that overweight/obesity was not associated with a high risk for CKD in subjects with MAFLD, irrespective of diabetes. Different study populations (population-based or hospital cohorts), types of studies (cross-sectional or longitudinal), and adjusted factors may explain the difference in results. Previous studies have indicated that the highest prevalence of lean NAFLD was found in Asian individuals (*Das et al., 2010*), and compared to whites with similar BMI values, Asians exhibit a higher proportion of visceral fat and lower lean body mass. Thus, there may be important differences among individuals of Asian and non-Asian ethnicity, this may be related to BMI cutoff values, lifestyle, dietary customs, and gut microbiota (*Yun et al., 2019*). However, when studies were stratified by region, no differences of the risk of developing CKD were found in distinct region.

Numerous evidence suggests that NAFLD is strongly correlated with the progression of CKD (*Targher et al., 2010*; *Targher, Chonchol & Byrne, 2014*; *Vilar-Gomez et al., 2017*). Due to the higher metabolic risk in obese NAFLD patients, it would be easy to predict that they are at higher risk to develop CKD. However, the impact of non-obese NAFLD on negative renal outcomes has not yet been adequately explored. Non-obese NAFLD patients had a high prevalence of metabolic syndrome. CKD may be the result of the adverse effects of the metabolic syndrome such as insulin resistance. Insulin resistance can lead to the activation of the renin–angiotensin system, which is key driver of renal damage. Furthermore, the steatotic and inflammatory liver itself has been known to be responsible for kidney injury in NAFLD (*Hydes et al., 2020*). Steatohepatitis stimulates the synthesis of inflammatory mediators such as cytokines, reactive oxygen species, and lipopolysaccharides, aggravating insulin resistance, tissue inflammation, and endothelial damage (*Dogru et al., 2013*; *Smith et al., 2012*; *Mima et al., 2018*). Lean patients with NAFLD are often referred to as metabolically obese normal weight, metabolically unhealthy status showed a greater risk for NASH and advanced fibrosis (*Phipps & Wattacheril, 2020*). Non-obese patients with metabolic risk factors suffered severe liver damage than those obese with healthy profile. *Ampuero et al. (2018)* revealed non-obesity NAFLD with unhealthy metabolism displayed a higher cardiovascular risk than obese with a normal metabolic condition, irrespective of the presence of obesity, reinforcing the importance of the metabolism beyond obesity.

To our knowledge, there are limited studies to assess the risk of CKD in non-obese NAFLD and obese NAFLD. The study comprehensively evaluated the risk of kidney damage in both populations. In addition to assessing the risk of developing CKD, we also compared eGFR and serum creatinine levels, which are key indicators of kidney function. However, several limitations were presented in our study. First, there was heterogeneity in our study. We attempted to analyze the sources of heterogeneity by subgroup analysis. Gender, the severity of NAFLD, hypertension, metabolic status, and diabetes could be the causes of heterogeneity, but the original data may not be obtained. Second, some studies reported incomplete adjustments or not adjusting for established risk factors and potential confounding variables, the real risk between the non-obese and obese NAFLD should be interpreted cautiously. Third, non-obese NAFLD patients showed a greater rate of PNPLA3 rs738409 mutations than obese NAFLD patients (*Wei et al., 2015*). PNPLA3 rs738409 are also an independent risk factor for CKD (*Liu et al., 2023*). Effect of genetic variants on the association between NAFLD and CKD cannot be assessed. Last, changes of BMI prior to baseline or during follow-up may influence our results.

## CONCLUSIONS

In summary, our meta-analysis demonstrated that non-obese NAFLD was not related to an increased risk of incident CKD compared with obese NAFLD. After studies were stratified by BMI cut-off values, diagnostic methods, regions, type of article, and article quality, the finding remained consistent. However, larger clinical cohorts with comprehensive adjustments for potential covariates are needed to better assess the risk.

### Funding

This work was supported by the National Key Research and Development Program of China (grant numbers 2022YFC2304800), and National Natural Science Foundation of China (grant numbers 82172254). The APC was funded by the Science and Technological Supports Project of Sichuan Province, China; No. 2024YFFK0214. The funders had no role in study design, data collection and analysis, decision to publish, or preparation of the manuscript.

### Grant Disclosures

The following grant information was disclosed by the authors:
National Key Research and Development Program of China: 2022YFC2304800.
National Natural Science Foundation of China: 82172254.
Science and Technological Supports Project of Sichuan Province: 2024YFFK0214.

### Competing Interests

The authors declare that they have no competing interests.

### Author Contributions

- Yixian You performed the experiments, analyzed the data, authored or reviewed drafts of the article, and approved the final draft.
- Xiong Pei performed the experiments, analyzed the data, authored or reviewed drafts of the article, and approved the final draft.
- Wei Jiang analyzed the data, prepared figures and/or tables, and approved the final draft.
- Qingmin Zeng analyzed the data, prepared figures and/or tables, and approved the final draft.
- Lang Bai analyzed the data, prepared figures and/or tables, and approved the final draft.
- Taoyou Zhou analyzed the data, prepared figures and/or tables, authored or reviewed drafts of the article, and approved the final draft.
- Xiaoju Lv analyzed the data, authored or reviewed drafts of the article, and approved the final draft.
- Hong Tang conceived and designed the experiments, authored or reviewed drafts of the article, and approved the final draft.
- Dongbo Wu conceived and designed the experiments, authored or reviewed drafts of the article, and approved the final draft.

### Data Availability

This is a systematic review/meta-analysis.

### Supplemental Information

Supplemental information for this article can be found online at http://dx.doi.org/10.7717/peerj.18459#supplemental-information.

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
