# Peer review of "Non-obese non-alcoholic fatty liver disease and the risk of chronic kidney disease: a systematic review and meta-analysis"

_PeerJ, doi:10.7717/peerj.18459_

## Round 0.1 · original submission · Major Revisions

The reviewers have highlighted several areas for improvement, with specific reference to the experimental design.

Reviewer 1 ·

Basic reporting

no comment

Experimental design

no comment

Validity of the findings

no comment

Additional comments

The authors conducted a meta-analysis of the association between NAFLD and CKD. The topic is very intriguing and important. These are my comments.

1. The association of NAFLD with CKD may involve inflammation. Please cite and discuss the following paper.

Obesity‐associated glomerular inflammation increases albuminuria without renal histological changes
FEBS open bio 8 (4), 664-670

Implications of treatment that target protective mechanisms against diabetic nephropathy
Seminars in nephrology 32 (5), 471-478

2. The association between NAFLD and fib-4 index is known, while the relationship between fib-4 index and CKD has been clarified. Please cite and discuss the following paper.

Prediction of decreased estimated glomerular filtration rate using liver fibrosis markers: a renal biopsy-based study
Scientific Reports 12 (1), 17630



Reviewer 2 ·

Basic reporting

The manuscript "non-obese non-alcoholic fatty liver disease and the risk of chronic kidney disease: a
systemic review and meta-analysis" presents an analysis of 15 studies assessing the incidence of
chronic kidney disease in patients with non-alcoholic fatty liver disease.
The authors put a lot of work into the results. However, I have a few comments on the manuscript.
The abstract and summary are written correctly and very interestingly.
There is a mistake in the introduction, line 46, which should be corrected: "chronic liver disease
(CKD)..." CKD is an abbreviation for chronic kidney disease, please explain this abbreviation
elsewhere.
The methods and results are described in a very extensive way, but I would organize them in simple
tables or figures to make them more readable.
eGFR and creatinine concentration should not be divided into separate paragraphs, because one
results from the other using the CKD-EPI formula.
The discussion needs to be improved, too much content contains repetition of results and data not
related to the topic of the manuscript. The sentences are not consistent with each other.
The sentence in line 285-287 "our study comprehensively assessed the risk of kidney damage in those
two populations, and we not only assessed the risk of developing CKD, but also compared the eGFR,
and serum creatinine, which reflected the kidney function" is incomprehensible.
The sentence in line 270 "CKD is closely related to the metabolic syndrome...." is not entirely true, I
propose to leave only the second part of the sentence "...CKD may be the result of the adverse effects
of the metabolic syndrome....".
The authors did not propose any explanation for the results of the study, why the risk of chronic
kidney disease in both obese and non-obese is the same.

Experimental design

the manuscript is a literature review that has been designed in a correct manner

Validity of the findings

no comment

Reviewer 3 ·

Basic reporting

1. Your manuscript requires thorough editing, with a particular focus on English grammar, spelling, and sentence structure. Specific lines needing attention include 83, 90-91, 120-121, 126-127, 146, 185-186, 202, 220, 226, as well as Figure 2. I recommend having a colleague who is proficient in English and knowledgeable in the subject matter review your manuscript, or alternatively, engaging a professional editing service to ensure clarity and accuracy.

Experimental design

2.1 In 2020 and 2023, there has been a redefinition of NAFLD to metabolic dysfunction-associated fatty liver disease (MAFLD) and metabolic dysfunction-associated steatotic liver disease (MASLD)(Eslam et al.; Rinella et al.). The alteration in nomenclature has led to a reevaluation of epidemiological trends and their associations with incident CKD. Pennisi, G et al. reported that patients with MAFLD had higher rates of developing CKD compared to those with NAFLD(Pennisi et al.). The development of CKD in individuals with NAFLD/MASLD is likely driven by a complex interplay of metabolic and hemodynamic changes, lipid nephrotoxicity, and genetic predisposition. Therefore, in NAFLD/MASLD patients, it may be suggested that BMI may not be a significant risk factor for CKD.
2.2 I commend the authors for their comprehensive dataset, which includes PubMed, Scopus, and Web of Science databases. However, the literature review could be further enhanced by incorporating additional sources. I recommend supplementing the search with resources from the Cochrane Library and Embase to ensure a more exhaustive review of relevant studies.
2.3 The methods section requires further refinement. According to Kjaergaard, M et al, fibrosis-4 index (FIB-4) is a biomarker of fibrosis(Kjaergaard et al.), thus, it is inappropriate to use it as an inclusion criterion for NAFLD. Additionally, the sixth criterion listed in the inclusion criteria (lines 94-96) should be reassigned to the exclusion criteria.

Validity of the findings

3.1 The primary focus of your study is to evaluate the incidence of Chronic Kidney Disease (CKD) in individuals with Non-alcoholic Fatty Liver Disease (NAFLD), differentiating between those who are non-obese and those who are obese. However, existing meta-analyses from 2014(Musso et al.), 2018(Mantovani, Zaza, et al.) and 2022(Mantovani, Petracca, et al.) indicate that NAFLD is associated with an increased risk of incident CKD independent of obesity. Specifically, a meta-regression analysis conducted by Mantovani et al. found no significant impact of body mass index (BMI) on the relationship between NAFLD and CKD risk(Mantovani, Petracca, et al.). Given this context, the significance of your study’s findings appears to be underexplored. Please elaborate on your rationale for selecting this topic and clarify the innovative contributions of your research.

Additional comments

4. Several references in this article should be updated. Incorporating high-quality literature from the past five years would enhance the articles’ relevance and reflect the most current advancements in the field.

---

## Round 0.2 · accepted · Accept

The reviewers are satisfied with the amendments to the manuscript and reason it ready for publication.

Reviewer 1 ·

Basic reporting

no comment

Experimental design

no comment

Validity of the findings

no comment

Additional comments

The manuscript is worth to be published.